# Very High Density Point Clouds from UAV Laser Scanning for Automatic Tree Stem Detection and Direct Diameter Measurement

**Karel Kuželka *** , **Martin Slavík and Peter Surový**

Faculty of Forestry and Wood Sciences, Czech University of Life Sciences, Prague, Kamýcká 129, 165 00 Praha, Czech Republic; mslavik@fld.czu.cz (M.S.); surovy@fld.czu.cz (P.S.)

* Correspondence: kuzelka@fld.czu.cz; Tel.: +42-022-348-3796

**Abstract:** Three-dimensional light detection and ranging (LiDAR) point clouds acquired from unmanned aerial vehicles (UAVs) represent a relatively new type of remotely sensed data. Point cloud density of thousands of points per square meter with survey-grade accuracy makes the UAV laser scanning (ULS) a very suitable tool for detailed mapping of forest environment. We used RIEGL VUX-SYS to scan forest stands of Norway spruce and Scots pine, the two most important economic species of central European forests, and evaluated the suitability of point clouds for individual tree stem detection and stem diameter estimation in a fully automated workflow. We segmented tree stems based on point densities in voxels in subcanopy space and applied three methods of robust circle fitting to fit cross-sections along the stems: (1) Hough transform; (2) random sample consensus (RANSAC); and (3) robust least trimmed squares (RLTS). We detected correctly 99% and 100% of all trees in research plots for spruce and pine, respectively, and were able to estimate diameters for 99% of spruces and 98% of pines with mean bias error of −0.1 cm (−1%) and RMSE of 6.0 cm (19%), using the best performing method, RTLS. Hough transform was not able to fit perimeters in unfiltered and often incomplete point representations of cross-sections. In general, RLTS performed slightly better than RANSAC, having both higher stem detection success rate and lower error in diameter estimation. Better performance of RLTS was more pronounced in complicated situations, such as incomplete and noisy point structures, while for high-quality point representations, RANSAC provided slightly better results.

**Keywords:** UAV; LiDAR; forestry; tree detection; diameter estimation; DBH; RANSAC; robust fitting

## 1. Introduction

Recent sustainable forestry standards require careful planning based on highly accurate inventory data of forest stands and properties [1]. Increasing demands on inventory data quality together with increasing costs of human labor in advanced countries force the forest owners to increase the efficiency of data collection and to simplify assessment of required parameters of forest trees and stands in means of automation. In the last decade, special attention has been paid to noncontact data collection methods providing accurate three-dimensional data that allow reconstructing forest stands and effectively estimating their parameters. The novel methods, made possible by advances in technology and computer vision algorithms, are mostly represented by two technologies: laser scanning and multiview photogrammetry.

Laser scanning methods utilize light detection and ranging (LiDAR) technology for precise range measurement of objects in surroundings. The distance calculation is based either on measuring the time needed for a light pulse to travel the distance to the target and back to the sensor, or on measuring

the wave phase of the reflected beam with known wavelength. As result, laser scanners provide 3D positions of up to 1 million points per second with a millimeter level precision.

Multiview photogrammetry is a more recent approach, triggered by recent advances in computer vision and development of algorithms like scale-invariant feature transform (SIFT) [2]. It reconstructs 3D surfaces of objects by calculating 3D positions of identical features identified in subsequent images in image sequences acquired with digital cameras [3].

Although both approaches result in 3D point clouds representing the surface of terrain and other objects, such as trees, due to different nature of data origin, LiDAR and photogrammetric clouds embody different properties. Regarding forest stand reconstruction, photogrammetric data are able to precisely represent the upper canopy envelope, whereas LiDAR data show greater capability to penetrate through canopies [4]. Although canopy height models can be accurately calculated from both types of data [5–7], LiDAR data show markedly superior accuracy for forest structure reconstruction and detailed terrain mapping [8]. During the last decade, LiDAR data were acquired particularly in two ways, aerial laser scanning (ALS) and terrestrial laser scanning (TLS).

ALS usually covers areas on regional scale. High density ALS (10 pulses/m$^2$ and more) provides sufficient detail to detect individual trees either from ALS-derived canopy height models [9] or the whole depth of ALS data [10], but typically, parameters of forest stands are derived from ALS data using an area-based approach that considers area as the unit of interest and generally includes estimates of forest variables on stand-level, per hectare values or mean tree attributes [11]. Beyond estimating basic forest parameters such as diameters, height, volume [12–14], biomass or biomass change [15,16], ALS data can be related to leaf area index [17], gap fraction, defoliation [18] or site index [19]. Nowadays, approaches for predicting tree diameter distributions from ALS data have emerged [8,20], helping to bridge the gap between area-based and tree-based inventories.

On the contrary, TLS provides ultra-high-density scans for detailed and accurate reconstruction of a forest stand, allowing for deriving virtually any dimension of the forest trees, but produces plot-wise data of low spatial extent and is time- and labor-demanding. As shown by numbers of studies, TLS point clouds allow for automatic detection and mapping of forest trees [21] and estimation of diameters [22] or the complete stem curve and tree architecture [23,24]. Additionally, TLS point clouds allow the determination of accurate nondestructive estimates of aboveground biomass [25] and even temporal development of aboveground biomass [26].

LiDAR data acquired from UAS platforms (ULS) represents a revolutionary type of 3D LiDAR data that joins benefits of both ALS and TLS. Multireturn lightweight laser scanners designated for unmanned aerial vehicle (UAV) carriers can reach a measurement rate of up to hundreds of thousands of measurements per second with the presented distance measurement accuracy/precision of 10 mm/5 mm [27]. Due to the low flying altitude, varying around 100 m above ground level, and arbitrarily low speed of multicopter-type UAV carriers, the density of resulting point clouds can reach the level of thousands of points per square meter. Such point clouds constitute a high-quality representation of 3D structure of forest stands and individual trees. Moreover, a relatively large range of scanning angles ensures a good coverage of ground, forest canopies and even individual tree stems.

The possibility of assessing parameters of individual tree stems in ULS data was recently tested by several authors. Brede et al. [28] showed that it was possible to detect stems and estimate diameters from ULS point clouds acquired with RIEGL VUX-1UAV scanner mounted on RiCOPTER. As they report, about two-thirds of detected stems were suitable to estimate diameter at breast height (DBH), while the DBH extraction workflow comprised a large share of manual work. The possibility of DBH extraction was also investigated by Wieser et al. [29]. They used the same scanning system—RiCOPTER and VUX1-UAV—to acquire ULS data of broadleaf forest in leaf-off condition, thus with very high density of points on stems. Their work was not focused on automatic tree detection; therefore, they manually selected the trunk sections in order to perform the cylindrical fit for DBH estimation.

Automatic workflows for stem recognition and DBH estimation have been published for TLS data. Olofsson et al. [30] successfully utilized random sample consensus (RANSAC) algorithm to

fit the circle on semicircular data representing one side of a stem. A similar problem was solved using Ganders' direct method for data from mobile laser scanning [31]. Other studies [22,32] suggest other algorithms for tree diameter estimation from dense TLS or photogrammetric 3D point clouds. However, the point density on stem surfaces in ULS data is significantly lower than that from terrestrial methods of data collection, and the noise shows markedly higher range. While the tree stem in terrestrial photogrammetric or laser point clouds may be represented by thousands points per meter of the stem length, the ULS data provide only tens of points, and the millimeter-level noise of TLS data grows to centimeter-level in ULS clouds. Moreover, parts of stems may be shaded by canopies, especially in leaf-on conditions or in coniferous stands. As a result, ULS data must be treated differently than TLS data, and the workflows must be modified to respect the ULS data specification.

This study evaluates the possibilities of detection of tree stems and diameter estimation using an automatic workflow in ULS data representing typical mature forest stands of two main economically important coniferous species of Central European forests—Norway spruce and Scots pine.

## 2. Materials and Methods

### 2.1. Research Area

For the study, the two most abundant and economically most important tree species of Central European forests were selected: (1) Norway spruce (*Picea abies* (L.) H. Karst.) and (2) Scots pine (*Pinus sylvestris* L.). The research area was located in the Central Bohemia region, Czech Republic, in two locations: (1) spruce-dominated upland (49.956N, 14.828E, 430 m AMSL) for Norway spruce, and (2) pine-dominated sandy area of the Czech basin (50.562N, 14.725E, 330 m AMSL) for Scots pine.

Pure-plantation forest stands were selected for both species: one large forest stand for Scots pine and two smaller stands for Norway spruce. All study stands were homogenous, mature, even-aged stands. The pine forest stand, as well as one of the spruce stands (plot 3) represented the typical structure of mature production forest stands of the given species. The other spruce stand (plots 1 and 2) represented a stand on very high quality site, which is illustrated by tree heights reaching 42 m and presence of dense understory. For validation purposes, six square research plots of the size 25 × 25 m were established, i.e., three for each species. Locations of sample plots and trees are illustrated in the height maps of forest stands in Figure 1. Table 1 describes parameters of forest stands and research plots. The inner structures of forest stands of both species are shown in Figure 2.

**Table 1.** Parameters of forest stands containing research plots.

| Plot ID | Species | Age (Years) | Density (Trees/ha) | Trees in Plot | Diameter (cm) | Mean Height (m) |
|---|---|---|---|---|---|---|
| 1 | Spruce | 130 | 400 | 24 | 29–57 | 39 |
| 2 | Spruce | 130 | 450 | 27 | 32–67 | 40 |
| 3 | Spruce | 110 | 320 | 20 | 15–46 | 31 |
| 4 | Pine | 100 | 160 | 8 | 27–35 | 24 |
| 5 | Pine | 100 | 360 | 22 | 23–37 | 26 |
| 6 | Pine | 100 | 330 | 21 | 22–35 | 25 |

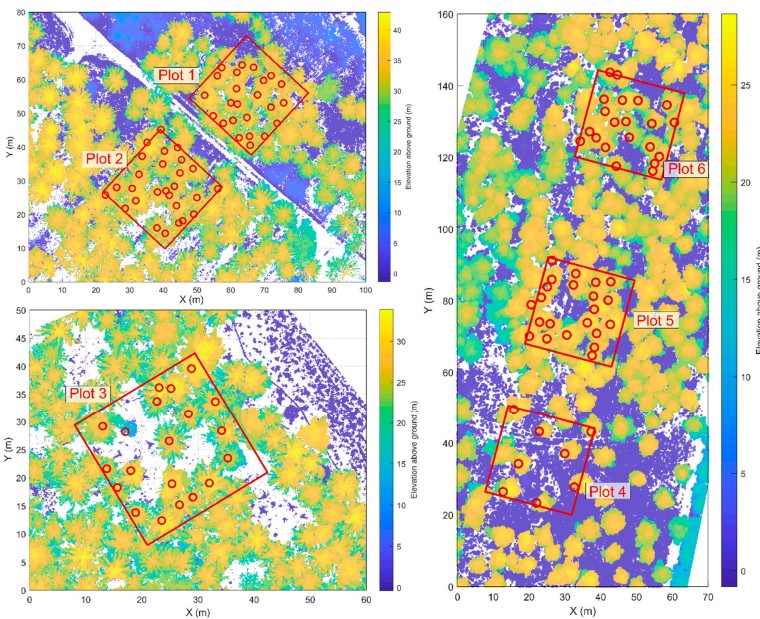

**Figure 1.** Location of six sample plots (red squares) and positions of measured trees (red circles). Colors in the maps express aboveground height according to the color bars.

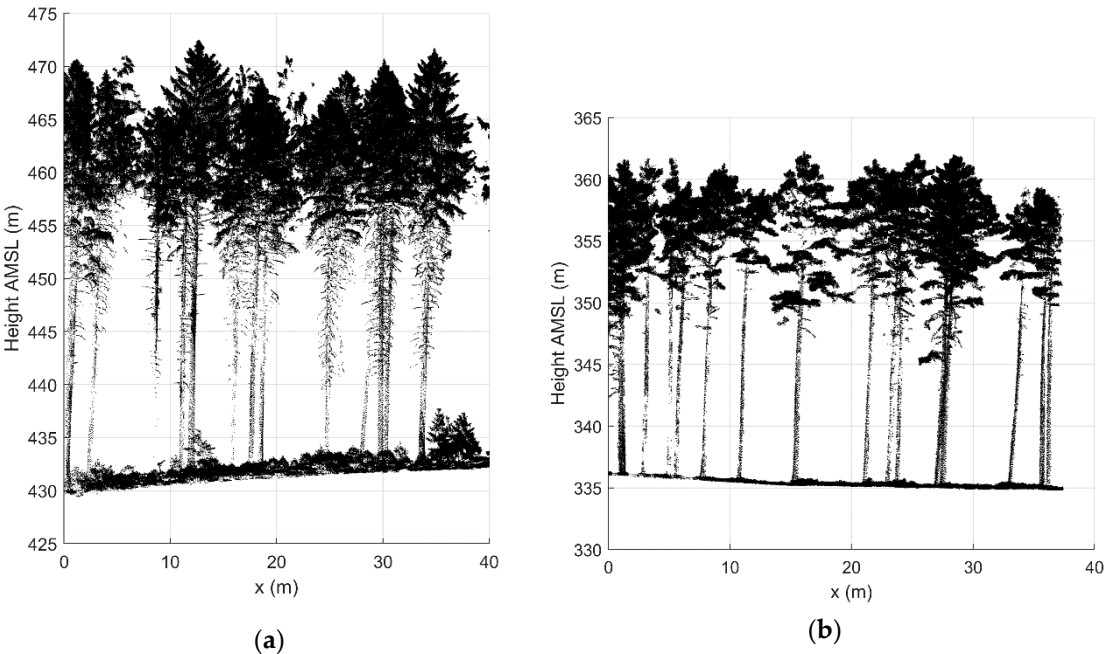

**Figure 2.** Structure of research forest stands: (**a**) a spruce stand (plot 1); (**b**) a pine stand (plot 5). Images show LiDAR representation of a transect 40 m long and 10 m wide.

## 2.2. Field Data

The research plots were located in the field using a Trimble C3 (Trimble Inc., Sunnyvale, California USA) total station. Positions of all trees in the square plots were measured and recorded with the total station. A tree position was defined as a center of the cross-section of the stem at 1.3 m above ground. To record the position of the true center of trees, cross-sections the angle offset measuring method was used: the cube corner prism was placed on the side of the tree preserving the distance to the center of the cross-section; subsequently, the angle to the tree center was recorded separately. The total station data were referenced using four RTK GNSS points. The final error of ground truth tree positions should not exceed 5 cm. Two diameters at breast height (DBH), i.e., 1.3 m above ground,

were measured in two perpendicular directions with a digital caliper Digitech Professional DPII (Haglöf Sweden AB, Långsele, Sweden), averaged by the caliper software and recorded as DBH for each tree.

## 2.3. ULS Data Acquisition

ULS data for the whole forest stands were acquired using a multirotor UAS RiCOPTER (RIEGL Laser Measurement System GmbH, Horn, Austria) equipped with RIEGL VUX-SYS laser scanning system. The VUX-SYS, consisting of VUX-1 UAV laser scanner, a GNSS and IMU system AP20 (Applanix: a Trimble company, Ontario, Canada) and a control unit, represents a complete lightweight solution for aerial laser scanning, primarily designated for the use with UAV platforms. VUX-1 UAV is a lightweight multireturn laser scanner with maximum laser pulse frequency of 550 kHz. For detailed description of the VUX-SYS and RiCOPTER, see [28].

For both locations, identical workflows for data acquisition were applied. Data were collected during an automatically executed flight over a predefined trajectory. The trajectory followed a double grid pattern, consisting of parallel lines with uniform spacing of 50 m, in two perpendicular directions, with stops and turns at vertices of the trajectory. The flight altitude was set to 80 m above ground, and the flight ground speed was set to 6 m s$^{-1}$. The mentioned flight parameters together with the laser pulse repetition rate of 550 kHz, forming 200 scan lines per second, correspond to average point density of 200 LiDAR points/m$^2$ for each single flight line. In order to ensure uniform point density and a regular pattern of the points, LiDAR data were collected during straight flight only, and data collection was interrupted while stopping and turning. LiDAR point clouds were generated from raw scan data in postprocessing that consisted of trajectory reconstruction with the use of postprocessing kinematic (PPK) method and Trimble VRS Now reference GNSS data in POSPac software (Applanix: a Trimble company, Ontario, Canada) and subsequent point cloud generation and adjustment with the use of RiPROCESS and RiPRECISION UAV software (RIEGL Laser Measurement System GmbH, Horn, Austria), following the official data processing workflow recommended by RIEGL. The classification of ground vs. nonground points was a part of the standard processing workflow in RiPROCESS. The final products of the processing were classified 3D point clouds stored in ASPRS LAS files.

## 2.4. Point Cloud Processing

Once the point clouds were produced, the environment of MATLAB R2017b (MathWorks Inc., Natick, Massachusetts, USA) was utilized to carry out analyses and point cloud processing and to generate graphic outputs. Firstly, thinning of the ULS point cloud was carried out to reduce the point cloud size in case of occurrence of areas with superfluous density and to unify the point distances. The minimum distances of the points were set to 0.01 m. To perform the thinning efficiently, the x, y, z coordinates of all points were rounded to the nearest 0.01 m. Subsequently, duplicities in coordinates were detected and removed. As the next step, noise filtering was applied. All points with a distance larger than 0.5 to the nearest group of points were considered as noise and removed. The next data processing comprised tree detection and segmentation and diameter estimation for detected trees.

## 2.5. Tree Segmentation

Methods for detecting trees in point clouds usually assume that tree stems are continuous solids reaching from ground to the canopy level or tree tops. Therefore the algorithms comprise vertical projections of points to the ground plate either as point counts in horizontal grid [30] or as bounding polygons of point clusters [10] in several horizontal layers, the width of which depends on point density. For ALS data [10], the layer width of 1 m was utilized, while the layer width for TLS data was set to approximately 3 cm [30] or 4 cm [22]. Wieser et al. [29] utilized 1 m layer width to detect the stems and fit cylinders to point structures in ULS data. Similarly, cylinder fitting based on RANSAC was applied to 0.5 m thick layers in detailed UAV-acquired photogrammetric point clouds of tree stems [33].

In this study, detection of individual trees was based solely on evaluating point densities in subcanopy space; the canopy height model (CHM), which serves as input data for several tree-segmentation algorithms, was not included in individual tree segmentation in this study. The whole subcanopy space was divided into voxels. The lower limit for the investigated subcanopy space was set to 0.5 m above ground to avoid low vegetation, and the upper limit was estimated as the average length of stems without branches; simultaneously, the limit was set so that the height of the investigated space was defined by whole number of meters. In our study, the upper limit was set to 9.5 m above ground. The voxel size was set according to the typical point density in ULS point clouds. As the tree stems were represented by 20–50 points per meter of stem length in average, horizontal voxel size was set to 0.5 m and the vertical size to 1 m. Counts of points were assessed in each voxel. Vertical projection of the voxel grid represented a horizontal raster that served for evaluating tree stem presence. A tree stem was more likely present in a raster cell when point counts in the corresponding voxels were high, continuous throughout the voxels and uniform. The raster cells were populated with values of stem presence indicator (SPI) that was formulated as:

$$\text{SPI}_{i,j} = \sum_{k=1}^{m-1} \sum_{l=k+1}^{m} n_{i,j,k} \cdot n_{i,j,l} \tag{1}$$

where $\text{SPI}_{i,j}$ is the value of the *i,j*-th cell of the raster; *m* is the number of voxels in vertical direction; and $n_{i,j,k}$ is the point count in the *i,j,k*-th voxel, where *i, j* and *k* are the voxel indices along x, y and z axis, respectively. This indicator prefers continuity and uniformity of point counts and gives provisions for higher point densities. High values of the indicator generally indicate locations where high point densities continue from ground to canopy base; therefore, tree stems can be discriminated from branches and other objects that are not continuous throughout the subcanopy space.

To detect local maxima in SPI raster, a maximum filter was applied and maximum raster $\text{SPI}_{max}$ was created. The value of *i,j*-th $\text{SPI}_{max}$ raster cell was defined as the maximum cell value in SPI raster of cells whose distance to the *i,j*-th cell was smaller than a defined radius. For a good result, the radius should be set as higher than the half distance between stems, but lower than the distance between stems. In our case, the radius was set to 2 m, according the a priori expected distance between trees. Local maxima in SPI raster are defined as cells where the SPI raster value equals the $\text{SPI}_{max}$ value, $\text{SPI}_{i,j} = \text{SPI}_{max(i,j)}$. To remove noise, a threshold for low SPI values was set. The threshold value corresponds to the approximate bottom limit of stem point distribution for subsequent diameter estimation and results from the requirement that the stem points are present and allow diameter estimation in a minimum of three sections. As the approximate minimum point count for efficient diameter estimation was expected to be 15, the appropriate threshold value corresponding to the requirement was 675. Detected local maxima in SPI raster with the value higher than the threshold were considered approximate stem positions.

To segment individual trees according to the detected positions, Delaunay triangulation was applied on the approximate stem positions, and the area of the forest stand was partitioned into polygons according to Voronoi diagram. Points in each polygon were subsequently treated as points belonging to one tree.

## 2.6. Diameter Estimation

Points representing a tree were partitioned in vertical sections according to their height above ground. With regard to the point densities (20–50 points per meter in average), height sections of 1 m were used for diameter estimation, because smaller sections do not provide enough points for reliable diameter estimation. Projection of points in height section to horizontal plane generated point representation of stem cross-sections. Diameters of the cross-sections were estimated as diameters of circles fitted in the point structures.

Point clouds were not manually cleared, and the cross-section point structures contained laser returns from branches, shrubs and other noise. Moreover, due to dense canopy and stems shading the laser beams, the cross-section representations might be poor or incomplete. Therefore direct least squares fitting techniques proposed in some studies [22,32] could not be utilized. Instead, methods allowing for circle detection in noisy and incomplete data were applied: (1) Hough transform, (2) RANSAC, and (3) robust least trimmed squares (RLTS).

Hough transform is a method originally developed to detect geometric objects in images, however the pixel-based algorithm can be adopted for application on point-based data. The Hough transform circle fitting is based on geometrical definition of a circle: a circle is a set of points with equal distance (i.e., radius) from the center. Therefore, if circles of a given radius are drawn around each point belonging to the circle perimeter, all the new circles will intersect in the center of the original searched circle. In practical application, a raster is established where each cell value represents the count of intersects of newly drawn circles with the given raster cell. A peak in the raster represents the center of the original circle. The weak point of the Hough transform is the need for a priori knowledge of the circle radius. If radius is unknown, the algorithm must be performed repeatedly with a set of different radii. The Hough transform is reported to be insensitive to noise or incompleteness of data [34]. Utilization of Hough transform for stem diameter estimation in TLS data was indicated by [35], together with detailed explanation of the method.

Random sample consensus (RANSAC) is an iterative stochastic method developed to fit a mathematical model in noisy data based on repeated model fitting to random subsamples. For circle fitting, a minimum subsample that defines a circle, i.e., 3 points, is repeatedly randomly selected, and the circle is fitted. The quality of each fit is evaluated by the ratio of points within a defined close distance from the fitted circle, so-called inliers. Finally, inliers of the best fit are used to fit the final circle. The number of iterations was set to 1000. Serviceability of RANSAC in forest mensuration was shown by [36], who utilized RANSAC for delineation of tree crowns in ALS data, and by [30] for diameter fitting in TLS clouds.

Robust least trimmed squares (RLTS) algorithm is a stochastic iterative method based on least squares criterion applied on a defined portion of smallest residues, which makes it more reliable in case of presence of outliers [37]. Analogous to RANSAC, RTLS iteratively fits circles to random subsamples of 3 points. In each iteration, squared residuals—squared distances of all points to the fitted circle—are calculated and sorted. A defined trim portion $h$ ($h > 1/2$) of points with the smallest residuals are selected and least square circle fitting is applied on the selected points. The criterion for evaluating the quality of fit is the sum of squared residuals of the least square circle fit. For detailed description of the method, see [37].

As shown by [30], the general algorithm can be adopted to better match the specific application of tree diameter estimation. Based on the assumption that a tree stem is a continuous solid along the vertical axis and with continuous diameter along the vertical axis, conditions on (1) counts of points inside the circle, (2) position of the circle and (3) radius of the circle can be set in order to eliminate circle fitting error.

The starting point for diameter fitting was determined from the vertical distribution of points as the local minimum in smoothed histogram of above-ground height. This way, the fitting started in a section that is above the shrubs and below the crown base. Optimally, this section contained mostly points belonging to stems and little noise; therefore, fitting should encounter the least problems and errors. After the starting section is fitted, fitting is carried out in the nearest lower section in successive steps. Fitting the subsequent sections can benefit from the known parameters of the previously fitted section.

For the starting section, the lower and upper limits for circle radius were set to 0.05 and 0.4 m as the most common tree radii in production forest; only circles with radii in this interval were searched in projections of section point clouds. If a circle was successfully fitted with radius $\hat{r}_i$ to the points

of section $i$, the following rules were applied for the radius limit $r_{\text{lim}}$ at the nearest lower $(i - 1)$th or nearest higher $(i + 1)$th section:

$$r_{\text{lim}(i-1)} = \{(0.8,\ 1.5) \cdot \hat{r}_i$$
$$r_{\text{lim}(i+1)} = (0.6,\ 1.2) \cdot \hat{r}_i \tag{2}$$

To limit the position of the circle in the $(i - 1)$th section, all points further than $2 \cdot \hat{r}_i$ from the center $[\hat{x}_i,\ \hat{y}_i]$ of the fitted circle in the $i$-th section were disabled. This action also eliminated the risk of false circle fitting in points representing returns from branches or shrubs. For all the circles, the limit count of points inside the circle was set to 25% of the count of perimeter points. Perimeter points were defined as points not further than 0.02 m from the circle perimeter; inside points were points inside the circle with distance to the circle perimeter higher than 0.02 m. For RLTS, the trim portion $h$ was set to two-thirds, i.e., 0.67.

Functions for all three circle fitting methods were written in the MATLAB environment as the general algorithms briefly described in this section and the mentioned adoptions of the algorithms were applied.

### 2.7. Accuracy Evaluation

Circles were fitted and diameters were estimated for all sections throughout the stem profile. Evaluation of diameter estimation was carried out for diameters at breast height. Diameters of second sections (aboveground height 1–2 m) were compared to field-measured DBH.

Errors of DBH estimations were assessed as a difference of estimated ($\hat{d}_i$) and field-measured ($d_i$) DBH. Relative error of DBH estimation was defined as DBH estimation error divided by field-measured DBH. At each plot, mean bias error (bias) and root-mean-square error (RMSE) were calculated both from absolute (cm) and relative (%) errors of $n$ trees detected and measured in the plot with each method. Mean bias error expressed the systematic error of diameter estimations in the plot; RMSE illustrated the typical extent of estimation errors.

$$\text{bias} = \frac{1}{n} \cdot \sum_{i=1}^{n} \left( \hat{d}_i - d_i \right)$$
$$\text{RMSE} = \sqrt{\frac{1}{n} \cdot \sum_{i=1}^{n} \left( \hat{d}_i - d_i \right)^2} \tag{3}$$

Position error of tree detection was calculated as a mean distance between detected and field-measured positions of all detected trees in a plot.

Statistical testing was involved in order to reveal factors influencing the accuracy of diameter estimation. Two-way ANOVA was utilized to investigate the influence of circle fitting method and tree species on absolute diameter errors. We also investigated the influence of point counts available for circle fitting on diameter estimation accuracy; to eliminate the effect of the previously mentioned factors, generalized linear model (GLM) was fitted with two categorical (circle fitting method and tree species) predictors and one quantitative (point counts) predictor. Linear regression models were built to quantify the effect of point counts for each fitting method separately.

## 3. Results

### 3.1. Point Clouds

The results of laser scanning and laser data processing are 3D point clouds consisting of approximately 2000 points per square meter (Figure 3). In spruce forest, approximately one-quarter of the total count of LiDAR returns was reflected from terrain surface, and the other three-quarters represent returns from vegetation. In pine forest stand, the ratio of ground vs. vegetation returns was

higher due to less dense crowns of pines compared to spruce; ground points represented approximately one-half of the whole amount of returns.

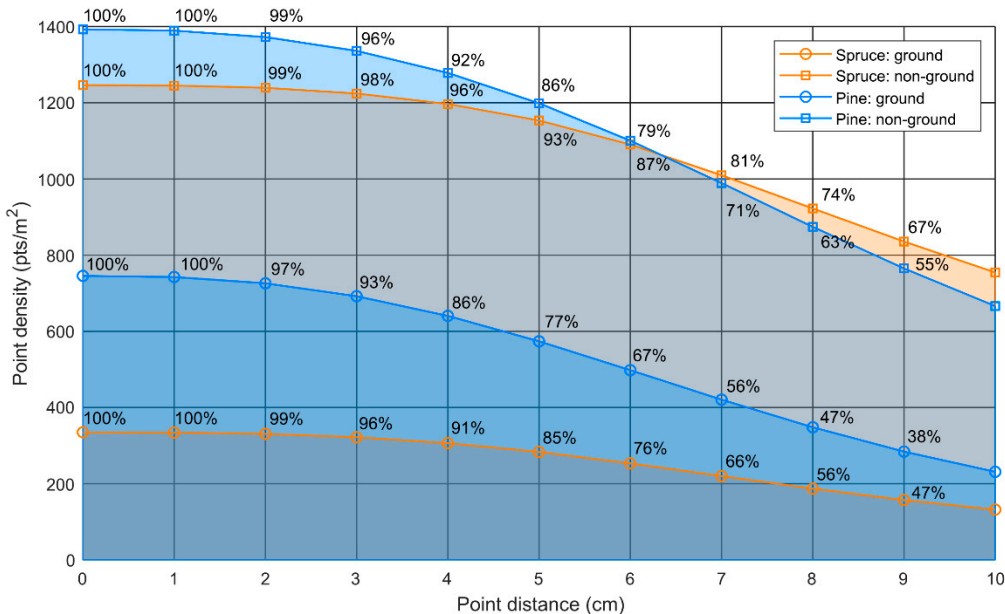

**Figure 3.** Gradual decline of point densities for voxel-based thinning with increased thinning intensity, i.e., point distances in resulting thinned point clouds. The densities show neglectable drop for thinning in voxels up to 2 cm.

The points were evenly distributed in 3D space, as illustrated by Figure 3, which shows point density and ratio of resulting point cloud size related to original point counts for different levels of thinning. For 1 or 2 cm thinning, the resulting point cloud contained almost 100% of points of original point clouds. For higher levels of thinning, the point counts in resulting clouds decreased. Well-covered surfaces contained points with regular distance of about 2 cm. For further processing, the point clouds were thinned to eliminate points with distances lower than 1 cm. Apparently, almost 100% of the information contained by the original point cloud remained in the thinned point cloud.

Vertical distribution of point densities in forest stands of both species is shown in Figure 4. Only points classified as nonground are displayed here. For both species, highest point densities are in the canopy space, reaching up to 180 points/m$^2$ in 1 m thick layer for spruce and >200 points/m$^2$ for pine. The lowest densities are found in the subcanopy space filled only by stems without branches. The average point densities of this space were below 3 points/m$^2$ both in spruce and pine research plots.

Regarding spruce stands, plot 3 shows a markedly different histogram from plots 1 and 2 due to significantly lower heights of trees in plot 3. Among pine plots, plot 4 shows significantly lower point densities than plots 5 and 6 due to lower canopy closure. Unlike pine, spruce stand (plots 1 and 2) shows increased densities in low above-ground layers (up to 5 m). These points represent returns from understory vegetation.

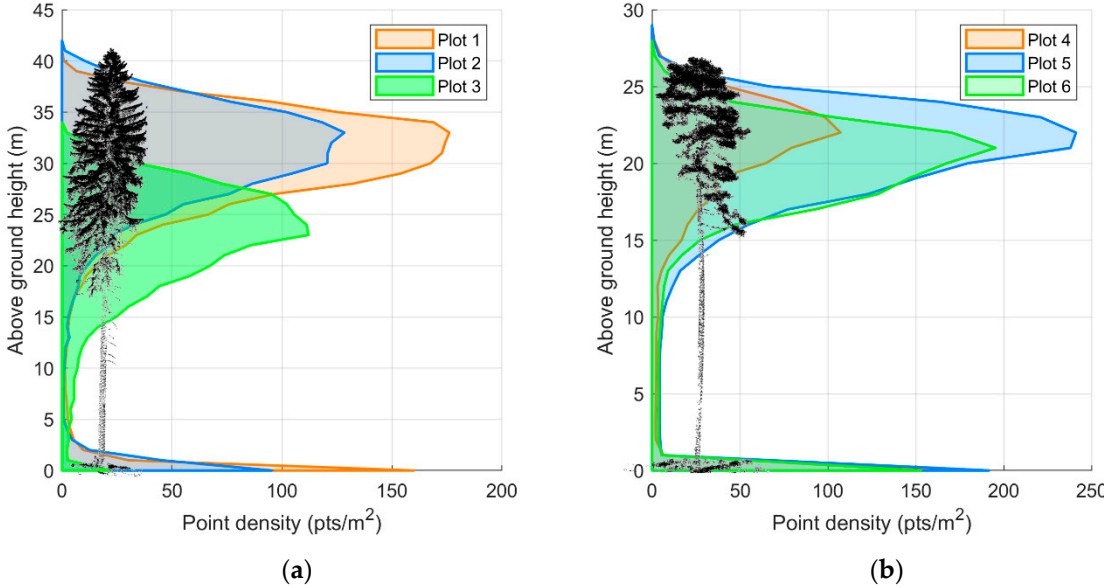

(**a**)　　　　　　　　　　　　　　　　(**b**)

**Figure 4.** Point densities in spruce (**a**) and pine (**b**) plots. Histograms represent average point counts per square meter in 1 m thick horizontal layers; only points classified as nonground are displayed.

## 3.2. Tree Segmentation

In pine research plots (plots 4–6), the automatic tree detection was fully successful. Pines are characterized by sparse crowns that allow a large portion of laser beams to penetrate under canopy and to return from stem surface. Moreover, there was no understory in pine plots. As result, the subcanopy part of point cloud consisted of clearly identifiable, well-covered stems. All the stems were correctly identified, and there were no false detections (Table 2).

**Table 2.** Counts of detected trees and detection errors in research plots. Meaning of columns is as follows: Counts of detected trees in plot (Detected); counts of correctly detected trees within 1 m distance from the real positions (Correct); counts of trees that were not detected (Omission); counts of false detections (Commission); average distance of correctly detected trees to real positions (Distance).

| Plot ID | Detected | Correct | Omission | Commission | Distance (m) |
|---|---|---|---|---|---|
| 1 | 24 (100%) | 24 (100%) | 0 | 0 | 0.46 |
| 2 | 31 (115%) | 26 (96%) | 1 (4%) | 5 (19%) | 0.38 |
| 3 | 20 (100%) | 20 (100%) | 0 | 0 | 0.27 |
| 4 | 8 (100%) | 8 (100%) | 0 | 0 | 0.37 |
| 5 | 22 (100%) | 22 (100%) | 0 | 0 | 0.27 |
| 6 | 21 (100%) | 21 (100%) | 0 | 0 | 0.30 |

In spruce stands, most laser beams are captured by denser and longer crowns of spruce, and therefore some stem parts or whole stems can obscured and contain too few points to be detected in the subcanopy space. However, in plots 1 and 3, all stems were correctly identified with no false detections (Table 2). Plot 2 was placed in an exceptionally dense forest stand that prevented one of the 27 tree stems, which was located in the densest part of the plot, from being well covered by LiDAR returns. Moreover, dense understory vegetation consisting of young spruces up to 5 m tall resulted in five false detections in plot 2 (Figure 5). However, it should be mentioned that false detections (commission error) do not imply errors in final result. Objects responsible for false stem detections do not have circular cross-sections; therefore, circles cannot be fitted, and the false detections are eliminated in the next step of point cloud processing.

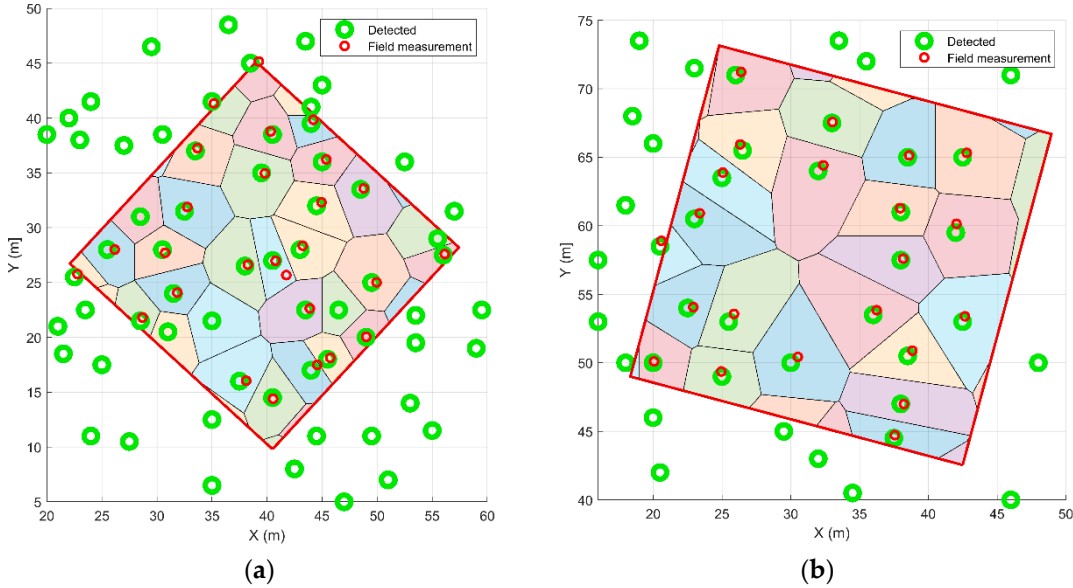

**Figure 5.** Tree positions automatically detected in point clouds and field-measured: (**a**) spruce plot (plot 2); (**b**) pine plot (plot 5). Colored polygons represent Voronoi diagram used to segment the plot area into regions of individual trees.

Tree positions recorded during tree segmentation represent centers of cells of the square grid, and the positions were recorded with 0.5 m resolution. Therefore, positions of detected trees do not correspond to field-measured positions (Figure 5). Distances indicated in Table 2 denote distance between real positions and cell center. The real distance between measured position and detected tree center is specified in the next step.

### 3.3. Diameter Estimation

Circles were fitted and diameters were estimated for all sections throughout the stem profile (Figure 6). As the figure indicates, circles representing stem cross-sections were well identified in most of the stem profile, even in crown sections containing branches and in bottom sections containing returns from understory vegetation. However, the presented results evaluate only quality of DBH estimation from the second above-ground section. Evaluation of estimated diameters of top sections is not presented here, as field-measured values were available only for DBH.

Not all tree stems automatically detected in the sample plots were represented by sufficient counts of LiDAR returns. Point representation of some trees did not allow for circle fitting in DBH layer due to low point density on the lower stem surface (details in Table 3). This applies especially to spruce trees; DBH of spruces could be quantified for approximately 90% of trees present in the plots, on average, with the RANSAC method. However, RLTS allowed to estimate diameters of 99% of all spruce trees present in the research plots. In pine forests, virtually all trees, with an exception of one tree in plot 5, allowed for DBH estimation.

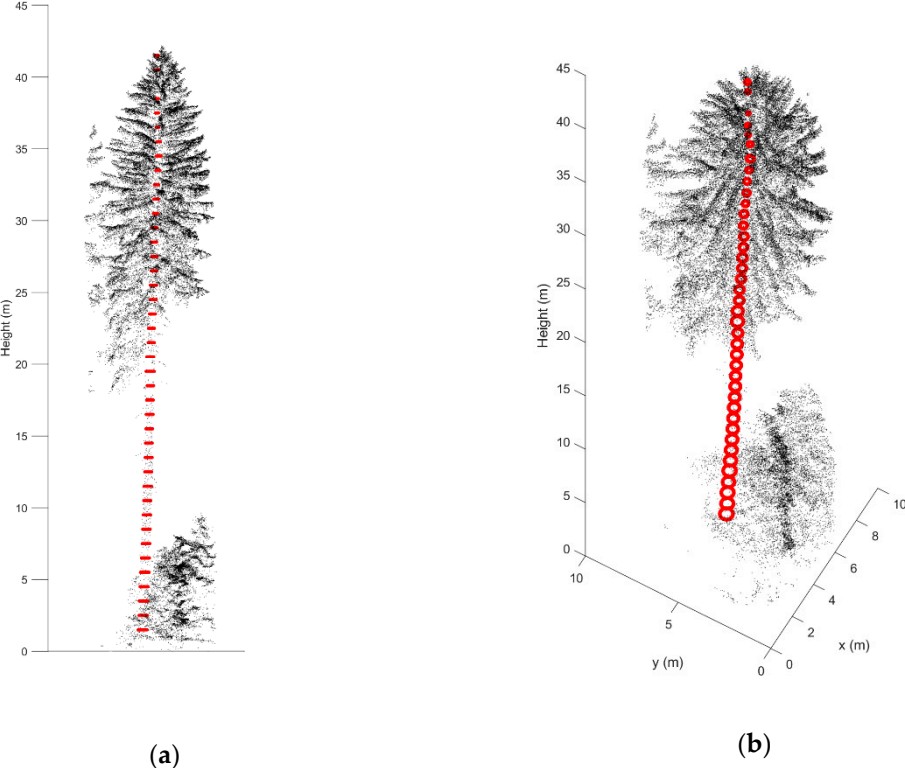

**Figure 6.** Automatically fitted perimeters of stem cross sections (red circles) in a point cloud representing automatically segmented spruce tree: (**a**) side view and (**b**) 3D view. The point cloud was thinned so that the fitted circles are visible in the figure.

The Hough transform method did not show the ability to fit circles to the point structures representing cross-sections of tree stems. In most cases, Hough transform did not find centers and radii of the circles, and the calculated radii were often determined by the radius limits (Figure 7). Therefore, the diameter estimation errors were high, usually with the highest RMSE among the three methods (Table 3).

**Table 3.** Results of diameter estimation. Table shows counts of trees that allowed for diameter at breast height (DBH) estimation and the respective percentage of present trees (Measured), mean error and RMSE of DBH estimation and mean positional error of detected trees (Position).

| Plot ID | Method | Measured | Mean Error (cm) | RMSE (cm) | Position (m) |
|---------|--------|----------|-----------------|-----------|--------------|
|         | Hough  | 21 (88%) | 5.0 (13%)       | 8.5 (25%) | 0.15         |
| 1       | RANSAC | 22 (92%) | 0.1 (0%)        | 9.5 (22%) | 0.13         |
|         | RLTS   | 24 (100%)| 0.7 (1%)        | 6.7 (17%) | 0.13         |
|         | Hough  | 19 (70%) | 1.0 (5%)        | 12.2 (28%)| 0.29         |
| 2       | RANSAC | 24 (89%) | 1.4 (4%)        | 4.2 (10%) | 0.15         |
|         | RLTS   | 26 (96%) | −1.1 (−3%)      | 6.2 (17%) | 0.15         |
|         | Hough  | 19 (95%) | 4.9 (14%)       | 9.0 (31%) | 0.25         |
| 3       | RANSAC | 20 (100%)| 3.2 (10%)       | 6.2 (22%) | 0.12         |
|         | RLTS   | 20 (100%)| 1.7 (6%)        | 5.9 (20%) | 0.12         |

**Table 3.** *Cont.*

| Plot ID | Method | Measured | Mean Error (cm) | RMSE (cm) | Position (m) |
|---|---|---|---|---|---|
| Spruce total | Hough | 59 (83%) | 3.7 (11%) | 10.0 (28%) | 0.23 |
| | RANSAC | 66 (93%) | 1.5 (5%) | 6.9 (18%) | 0.14 |
| | RLTS | 70 (99%) | 0.3 (1%) | 6.3 (18%) | 0.13 |
| 4 | Hough | 8 (100%) | −0.7 (−3%) | 7.8 (25%) | 0.13 |
| | RANSAC | 8 (100%) | 0.0 (0%) | 4.0 (12%) | 0.14 |
| | RLTS | 8 (100%) | −1.0 (−4%) | 4.4 (14%) | 0.15 |
| 5 | Hough | 21 (95%) | 1.7 (4%) | 10 (35%) | 0.18 |
| | RANSAC | 21 (95%) | 1.4 (4%) | 4.6 (14%) | 0.10 |
| | RLTS | 21 (95%) | 0.7 (−3%) | 4.3 (15%) | 0.10 |
| 6 | Hough | 17 (81%) | 7.8 (30%) | 15.1 (59%) | 0.10 |
| | RANSAC | 21 (100%) | 0.3 (2%) | 9.0 (31%) | 0.11 |
| | RLTS | 21 (100%) | 0.7 (−4%) | 7.1 (27%) | 0.12 |
| Pine total | Hough | 46 (90%) | 3.6 (12%) | 12 (44%) | 0.14 |
| | RANSAC | 50 (98%) | 0.5 (2%) | 7.0 (24%) | 0.11 |
| | RLTS | 50 (98%) | −0.8 (−3%) | 5.7 (21%) | 0.12 |
| All trees | Hough | 105 (86%) | 3.6 (11%) | 10.9 (36%) | 0.19 |
| | RANSAC | 116 (95%) | 1.2 (4%) | 6.8 (20%) | 0.13 |
| | RLTS | 120 (98%) | −0.1 (−1%) | 6.0 (19%) | 0.13 |

The other two algorithms, RANSAC and RLTS, provided much more reasonable results of circle fitting and diameter estimations (Figure 7). In general, RLTS performed slightly better than RANSAC (Table 3), having both higher ratio of measured trees and lower RMSE. In pine research plots, there was almost no difference in mean error (on average 2% and −3% for RANSAC and RLTS, respectively); RMSE of RANSAC (24%) was gently higher than that of RLTS (21%). The difference of the algorithms' performance appeared under the difficult conditions caused by incompleteness of point representations of cross-sections and presence of LiDAR returns from understory vegetation. In the plots with the highest errors (plots 1 and 6), RLTS showed about 2 cm lower RMSE in comparison with RANSAC; in plots with overall lower errors, RMSE of both methods were balanced.

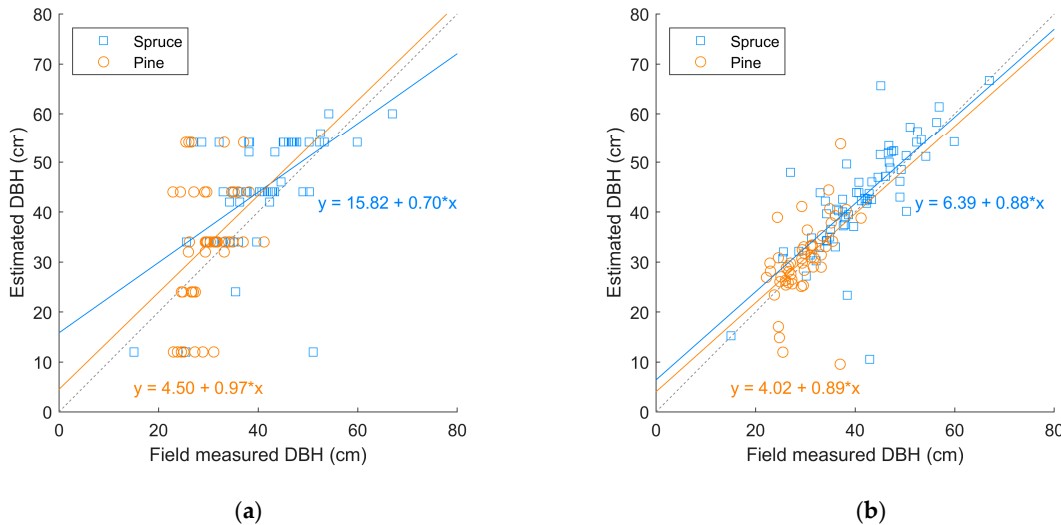

(**a**)　　　　　　　　　　　　　　　　　(**b**)

**Figure 7.** *Cont.*

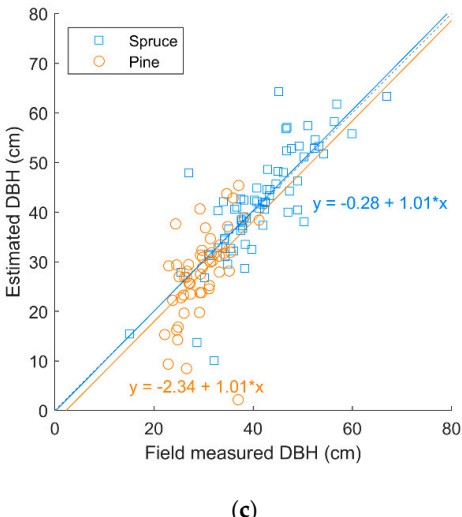

(**c**)

**Figure 7.** Field-measured vs. estimated diameters: (**a**) Hough transform; (**b**) RANSAC; (**c**) RLTS. Black dashed line expresses the 1:1 trend.

### 3.4. Factors Affecting the Accuracy of Diameter Estimation

The accuracy of diameter estimation is clearly dependent on the quality of point representation of stem cross-sections (Figure 8). Table 4 shows the dependency of DBH estimation error on point counts that represent the perimeter of stem cross-section. Higher point counts resulted in sections that were better represented, with better fit. As Table 4 indicates, sections with poor point representation were better fitted by RLTS; with increased quality of point representation, the difference between RANSAC and RLTS disappeared. For well-covered sections represented with more than 100 points, RANSAC provided better diameter estimation than RLTS.

Both methods (RANSAC, RLTS) showed significant trend (*p*-value < 0.05) in DBH estimation error with rising count of points in a section used to fit a circle (95% confidence bounds for slope parameter 0.001, 0.04 for RANSAC and 0.01, 0.05 for RLTS). While the error was negative (diameters were underestimated) for low counts of available points, diameters in sections represented by higher count of points were more likely estimated with positive error (diameters were overestimated).

Regarding the absolute error, a significant decline (*p*-value < 0.05) of diameter estimation error with rising point counts was observed with the RANSCAC method. More points representing the stem cross-section resulted in more accurate diameter estimation (95% confidence bounds for slope parameter −0.06, −0.002). No such trend was observed with the RLTS method.

Comparison of RANSAC and RLTS methods based on absolute errors showed that RLTS had lower error in comparison with RANSAC in sections where the overall error was high. On the contrary, RANSAC had lower errors in sections with low overall error. In other words, RLTS showed the ability to improve fit in more complicated situations, while RANSAC provided better estimations in sections with better quality of point description.

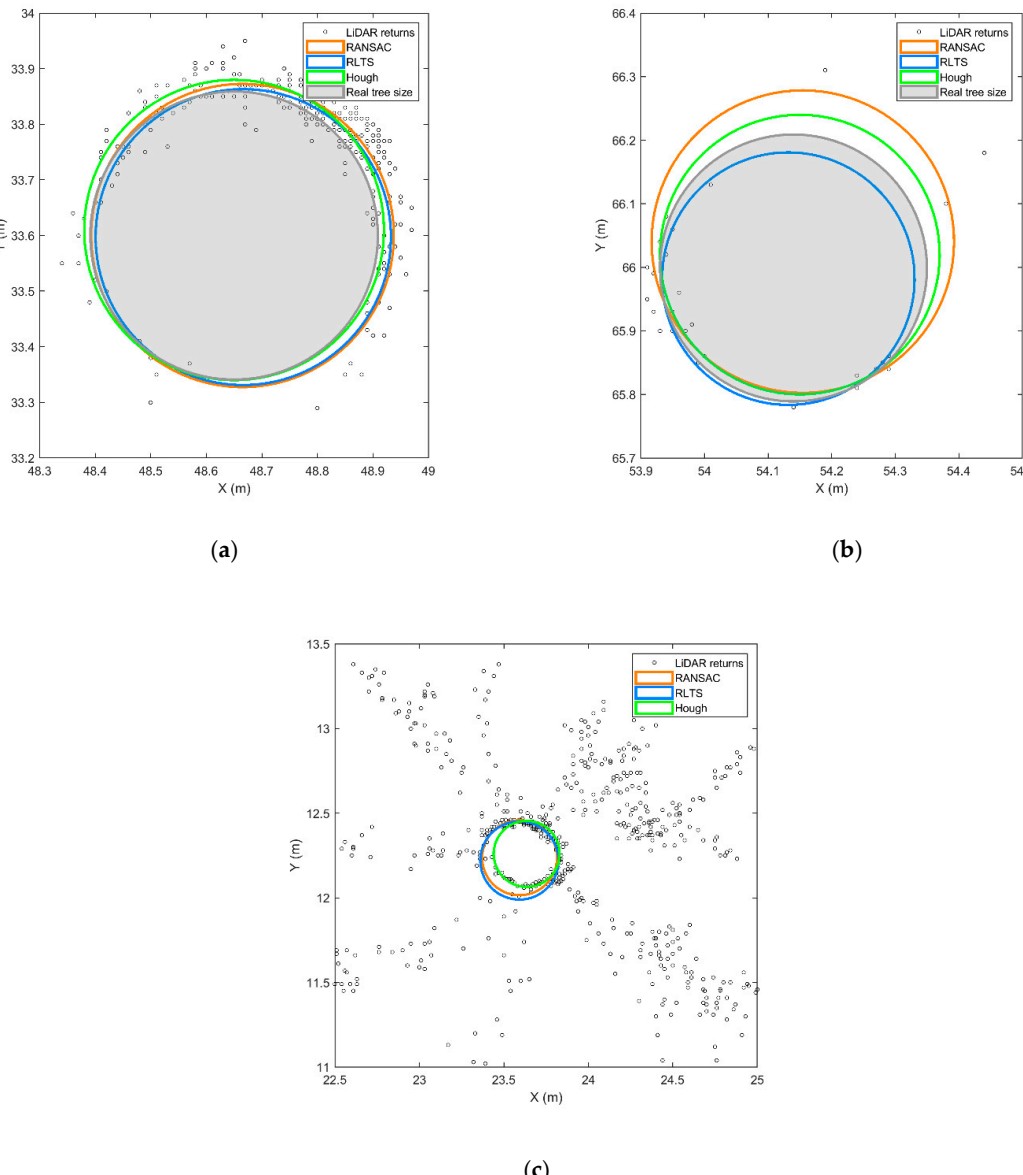

(a)                                                      (b)

(c)

**Figure 8.** Three examples of circle fitting in point representations of stem sections: (**a**) well covered stem section; (**b**) incomplete perimeter of stem section; (**c**) upper stem section with branches. The real diameter was not measured in (**c**).

**Table 4.** RMSE (cm) of diameter estimation for sections represented with different point counts.

| Point Count | Hough | RANSAC | RLTS |
|:---:|:---:|:---:|:---:|
| >5 | 11.3 | 6.8 | 6.0 |
| >10 | 10.9 | 6.1 | 5.6 |
| >50 | 10.2 | 5.5 | 5.7 |
| >100 | 10.4 | 4.7 | 4.8 |

Surprisingly, quality of diameter estimation was not influenced by species. Two-way ANOVA (significance level $\alpha = 0.05$) revealed the influence of fitting method (significant differences between Hough transform and the other two methods, $p = 0.02$ for RANSAC, $p = 0.001$ for RLTS; no difference between RANSAC and RLTS, $p = 0.95$) but no influence of species (no significant difference between spruce and pine, $p = 0.24$) on absolute errors of DBH estimate. GLM that tested the influence of (1) fitting method, (2) species and (3) point counts representing the cross-sections showed significant

influence of fitting method ($p = 0.014$) and point counts available ($p = 0.0015$) for circle fitting and again did not reveal any significant influence of species ($p = 0.81$). The point counts available for circle fitting were similar in both species (41 points for spruce and 47 points for pine, on average); the variance in point counts was double in spruce compared to pine.

The average position error of detected trees was 13 cm. This value comprises (1) positional error of acquired point cloud, (2) transformation error of point cloud transformation from WGS-84 coordinate system to the Czech official coordinate system S-JTSK (Krovak East North, EPSG:5514), (3) error of circle fitting methods in segmented point cloud and (4) error of field-measured coordinates of trees with the total station. Unfortunately, we were not able to quantify individual components of the positional errors.

## 4. Discussion

Three-dimensional LiDAR point clouds acquired from a UAV platform represent a relatively new type of remotely sensed data for forestry and environmental applications. Multireturn lightweight laser scanners designated for UAV carriers can reach a pulse rate more than half a million pulses per second with survey grade accuracy of 10 mm. Due to the low flying altitude, varying around 100 m above ground level, and arbitrarily low speed of multicopter-type UAV carriers, the density of resulting point clouds can reach the level of thousands of points per square meter. Such point clouds constitute a high-quality representation of 3D structure of forest stands and individual trees. In this study, we show that in ULS point clouds individual trees can be segmented and their dimensions measured in a fully automated process.

In our study, point clouds of forests of different species showed slightly different point cloud densities; this may be attributed to topographic effects. While the pine forest was in a flat area, spruce forest stands were located in a hilly area on a slope. As a result, above-ground flight height in a spruce area may vary during the flight, which may cause unequal point densities.

High density of canopy cover, together with understory, as present in plot 2, implicate challenging conditions both for data acquisition and automatic data processing. Moreover, as mentioned by Brede et al. [28], spruce, unlike broadleaved trees as beech and oak, do not allow for higher-order returns due to dense crowns with needles with high content of water. Therefore, we expected much lower success rate of tree detection in plot 2, characterized by tight canopy closure and trees as close as 1.4 m from each other in the densest part. However, with the exception of one single tree shaded by neighboring trees in the densest part, all stems were detected and even measured using RLTS algorithm.

RLTS was presented as an algorithm for robust cylinder fitting [37], but it can be utilized for circle fitting as well. It showed its ability, as declared in [37], to fit incomplete data as well as data containing noise and height percentage of outliers. In difficult conditions of incomplete or noisy data, RLTS performed better than RANSAC. Moreover, the diameters that were not estimated with RANSAC represent more complicated situations for circle fitting and increased the mean errors of RLTS. However, for well-covered tree sections, especially in low-density pine stand, RANSAC provided better estimations. This phenomenon may be caused by the fact that RLTS disables a defined portion of points with highest residuals. Removing noise points in noisy data improves the fit, but utilizing the information present in all points may lead to better estimation in high-quality point representation.

Hough transform was presented as a method for efficient circle fitting for diameter estimation in TLS data [35]. Due to the principle of Hough transform circle detection, point structure forming almost perfect circle or its part is required for the method to return correct results. Therefore, it might be a reasonable method for TLS data, which are characterized by ultra-high point density and low level of noise. ULS provides much lower densities, and LiDAR returns often form incomplete perimeters of cross-sections. Moreover, LiDAR data were not manually segmented or cleared, and the point cloud contained returns from other objects, like branches or understory vegetation. Under such conditions, Hough transform did not perform satisfactorily.

The previously published study of Wieser et al. [29], focusing on diameter estimation from manually delineated tree stems in ULS point cloud, was performed in a deciduous forest during leaf-off season. Thus, their point representation of stem surface was significantly denser. They did not provide RMSE of DBH estimation; however, their errors varied between −18 and 18 cm, with median absolute error of approximately 5 cm. Here, we show that comparable accuracy of DBH estimation can be reached even in coniferous forest with dense canopies and using automated segmentation. Brede et al. [28] report RMSE of diameter estimation 4.24 cm; however, their result is based on fitting 39 trees with the best point representation from the total 58 research trees. Moreover, the suitability for circle fitting was manually inspected for each tree, and outlier points, such as returns from branches, were manually removed. Comparably high RMSE of diameter estimation are reported from RANSAC diameter estimation from TLS data that provide significantly higher point densities and level of detail: Olofsson et al. [30] report RMSE of 33 to 59 mm for data without the removal of outliers. RMSE of 3 to 6 cm for diameter estimation is also reported from detailed photogrammetric reconstruction of forest stands, e.g., [32,38]. The results indicate that for the present we have to accept centimeter-level error for diameter estimation from all kinds of remotely sensed structural data.

The diameter estimation suffers from a relatively large dispersion of points around cross-sections' perimeters. The point dispersion is partially caused by the diameter change along the 1 m sections of tapered trees. However, the point density, which was around 44 points per meter of stem length in average, but sometimes as low as 5 points for the section, did not allow to fit circles in thinner layers. Thinner layers in many cases fail in circle fitting due to insufficient point representation of the circle. Due to the taper, the vertical point projection of a 1 m long section forms a point belt of the thickness of around 1–2 cm instead of a circle. Another component of the point dispersion is caused by LiDAR accuracy (1–2 cm) and bark structure (1–2 cm). These components represent an unavoidable error of the data acquisition. The most important source of point dispersion is the co-registration of scans from individual flight lines. Due to the limited accuracy of the utilized IMU, software-based co-registration of scans must be applied. Accurate co-registration is problematic in environments with complex structure, such as forests [28,29]. The co-registration error may reach several centimeters in forest environments, resulting in inconsistent and dispersed circle representation, visible in Figure 8.

In a single UAV flight, 10 to 20 hectares of forest area can be covered with LiDAR data. Considering time consumption for mission planning, flight preparation and transports between flight locations, the area that can be mapped in a day reaches 100 hectares. As a result, ULS is effective for detailed mapping at the level of forest stands or forest areas up to several hundreds of hectares. We showed that for mature, even-aged production forests, virtually every tree in the mapped area can be localized with an error that usually did not exceed the tree radius. The accuracy of diameter estimates may be sufficient for common stand inventories such as volume assessment before standing timber sale or forest management plan development. An important benefit of USL data is the possibility to measure upper stem diameters to reconstruct the taper curve for wood quality and assortment prediction. The biggest limitation of accurate diameter estimation—centimeter-level noise caused by inaccurate co-registration of scan lines—may limit circle detection in point clouds representing thinner trees. Revision of the methods or setting modifications may be needed in younger forest stands. The applicability of this method for tree detection and measurement in more complex stands, such as two-story stands is a subject for further research.

Most methods of individual tree detection from remotely sensed structural data rely on canopy height models [39,40] or point clouds of tree crowns [41,42]. In this study, we took the challenge of verifying whether tree detection and measurement can benefit from unrivaled density of point clouds acquired with ULS. Tree detection was based solely on points representing tree stems in subcanopy space, and diameters were assessed by direct measurement in acquired point clouds. However, stem points typically represent less than 1% of the total point counts in a vegetation point cloud, not considering the ground returns. Most points describe the structure of tree crowns and canopy. The success rate of tree detection would definitely be improved by involving segmentation

techniques based on structural information of canopy combined with stem detection proposed in this work. This combination would result in a complex method utilizing a higher portion of available information and evaluating the point cloud in its whole depth, in a similar way that was proposed by Ayrey et al. [10] for ALS data. We consider this as a direction of our future effort.

## 5. Conclusions

We evaluated the suitability of ULS data—laser scanning data acquired from an UAV platform—for individual tree detection and stem diameter estimation in a fully automated workflow. The study was performed in mature pure stands of the most abundant and economically important species, Norway spruce and Scots pine, in their typical growing conditions, which represent the typical situation of need for detailed forest stand inventory in practical management of production forests.

For both species, we were able to detect and localize 98–99% of all trees present in the research plots with an average positional error of 13 cm and estimate their diameters at breast height with bias of 0.1 cm (corresponding to 1% mean relative error) and with RMSE of 6 cm (corresponding to relative RMSE of 19%). We evaluated three algorithms for circle fitting in noisy data structures, Hough transform, random sample consensus (RANSAC) and robust least trimmed squares (RLTS). Hough transform is not a suitable method for diameter fitting in ULS data. RANSAC and RLTS can provide reasonable circle fit, whereas RLTS performs slightly better, especially in lower quality point representations of cross-sections; it shows both higher success rate and lower error.

**Author Contributions:** Conceptualization, K.K. and P.S.; methodology, K.K.; software, K.K.; validation, K.K. and P.S.; formal analysis, K.K.; investigation, K.K. and M.S.; resources, P.S.; data curation, K.K. and M.S.; writing—original draft preparation, K.K.; writing—review and editing, P.S. and M.S.; visualization, K.K.; supervision, P.S.; project administration, P.S.; funding acquisition, P.S. All authors have read and agreed to the published version of the manuscript.

**Funding:** This research was funded by OP RDE, grant number CZ.02.1.01/0.0/0.0/15_003/0000433 (Building up an excellent scientific team and its spatiotechnical background focused on mitigation of the impact of climatic changes to forests from the level of a gene to the level of a landscape at the FFWS CULS Prague) and Ministry of Agriculture of the Czech Republic grant number QK1920458

**Conflicts of Interest:** The authors declare no conflict of interest. The funders had no role in the design of the study; in the collection, analyses, or interpretation of data; in the writing of the manuscript, or in the decision to publish the results.

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
