# Peer review of "Very High Density Point Clouds from UAV Laser Scanning for Automatic Tree Stem Detection and Direct Diameter Measurement"

_remotesensing, doi:10.3390/rs12081236_

Round 1

Reviewer 1 Report

The paper presents a novel approach to use UAV laser scanning in automatic detection of trees of mature pine and spruce stands using stem detection instead of more common canopy height (CHM) segmentation -based method. Also, diameter at breast height (DBH) is estimated directly by automatic circle detection methods. The methods are presented earlier but not in combination and using automatic process. It is very nice to notice that the topic is focused on these two methods. It would have been very easy to add comparison with CHM method, tree height estimation etc. in the aims of the study. However, because of limited field data and novelty factor of the presented methods it is only good that these considerations were left out of the paper.

The structure of the paper is good. Some statements sound like marketing and language needs checking. Also, some parts of the methodology presentation need clarification and discussion part is missing comparison to earlier studies.  

Specific comments

Line 12: “…accuracy 10 mm…” Accuracy of 10 mm is presented by Riegl for the scanner system. It is a result from scanner manufacturer’s tests carried out in test conditions and is probably not valid in real world conditions in forest. Please use some more qualitative phrase to describe the accuracy.

Line 14: “Scotch” -> Scots. Same mistake everywhere in text.

Line 21: “mean error”? You men mean bias error (bias)?

Lines 23-24: “…having both higher success rate and lower error…” -> …having both higher stem detection success rate and lower error…”

Lines 31-32: “Recent sustainable forestry standards require careful planning based on highly accurate inventory data of forest stands and properties.” Any reference?

Line 56: Remove “as 1)”

Lines 65-66: Individual tree detection method based on high density lidar data was actually developed and published over 20 years ago and about at the same time with ABA method. Check for example Hyyppä, J and Inkinen I. 1999. Detecting and estimating attributes for single trees using laser scanner. Photogramm J Finland 16, 27-42 in comparison with Naesset’s paper published in Remote Sensing of Environment in 1997 and studies published by Nelson et all before that.

Line 77: “…with survey grade accuracy 10 mm.” Please give reference to this information.

Lines 83-107: Change the order of paragraphs so that at first tell what has been done earlier and after that what is the new thing in this paper and what are the specific aims.

Line 152: SI unit symbol is m s-1

Line 165: ULS data processing is described partly in previous chapter. This chapter is more about point cloud processing.

2.6. Diameter estimation

Please refine the presentation of the three different circle fitting methods. Describe which tools, libraries, software were used and give brief overall description of every method.

Line 218: “Trees presenting a tree” There is probably a mistake her?

Line 217- (chapter 217): It is not clear how DBH estimate is defined. Is it from observations between 1 and 2 meters above ground? If yes, how you justify that DBH is from this section? Why not use some other section or a curve fitted in sections’ heights and then exact height of 1.3 meters? Or is there something I did not understand?

Lines 266-270: Please provide formulas or references. It is not clear what authors mean by mean error.

Line 315: Is it really an exceptionally dense forest stand? I do not consider 450 stems/hectare as very dense.

Lines 318-321: If a detected tree without a successful diameter estimation are eliminated then the final tree detection rate is the outcome of estimated diameters. Not outcome of detected trees. Right?

Line 353-354: Of all correctly detected spruce trees. Not all spruce trees.

Figure 7: Image quality is very bad. Needs to be improved.

Lines 387-409: You provide here results for several statistical tests. Please mention these tests in methods section, too. Also, you could divide the chapter 3.3 Diameter estimation more clearly in presenting the estimation error and then the test results. Presenting test results (and doing the tests) is a good thing. The presentation of the test results could be improved.

Lines 415-416: The last sentence belongs to discussion part.

Discussion: Add findings of several previous studies about tree detection success rate and diameter estimation (TLS, ALS). While you present very good accuracy for tree detection rate the diameter estimation accuracy is not very convincing. Also, you could discuss is the diameter estimation accuracy good enough for practical use. Furthermore, how do you see your findings of your test case can be generalized? It seems that the forests present quite easy case for the presented method. Is the method applicable in multi species stands, stands with uneven age structure, young stands with much higher stem density…

Lines 490-491: “Considering that in a single UAV flight lasting 20 minutes, 10 to 20 hectares of forest area can be covered with high quality point cloud…” Maybe the flight time for data acquisition is about this in optimal conditions, but it requires that everything is ready for the data collection. It is more informative if you give your estimation on how many hectares can be acquired in a single day and how much time is required for planning before the data acquisition and analyzing the data after the data is acquired. Or you just can leave out this kind of marketing text.

Line 493: Should it be comparable?

Author Response

Dear reviewer,

Thank you for your valuable comments. We really appreciate your constructive effort that helped us to eliminate some mistakes in the manuscript, to improve the presentation of the message of our work, and to increase the understandability for readers.

Here your comments are listed together with our response to each comment.

Authors

Specific comments

Line 12: “…accuracy 10 mm…” Accuracy of 10 mm is presented by Riegl for the scanner system. It is a result from scanner manufacturer’s tests carried out in test conditions and is probably not valid in real world conditions in forest. Please use some more qualitative phrase to describe the accuracy.

In the abstract, we changed the expression “accuracy 10 mm” to a simple and more qualitative phrase “survey grade accuracy”. The presented values are referenced and explained in the introduction section: “…with the presented distance measurement accuracy/precision 10 mm/5 mm”. (Line 12)

Line 14: “Scotch” -> Scots. Same mistake everywhere in text.

All occurences in the text were corrected.

Line 21: “mean error”? You men mean bias error (bias)?

Yes, this means the bias error. It was specified in the text. (Line 21)

Lines 23-24: “…having both higher success rate and lower error…” -> …having both higher stem detection success rate and lower error…”

Modified according to the reviewer’s suggestion. (Line 24)

Lines 31-32: “Recent sustainable forestry standards require careful planning based on highly accurate inventory data of forest stands and properties.” Any reference?
The statement was referenced with FAO Voluntary guidelines on national forest monitoring; Rome, 2017 that mentions the need for detailed monitoring of forest sources (Evidence-based policies and practices that support highly productive and sustainably managed agricultural sectors are key to achieving FAO’s goals of eradicating hunger and eliminating poverty for the benefit of present and future generations. To achieve these goals, stronger national capacities to collect, compile and analyse data, and to generate and disseminate information tailored to specific audience needs are essential.) (Line 33)

Line 56: Remove “as 1)”

Modified according to the reviewer’s suggestion. (Line 57)

Lines 65-66: Individual tree detection method based on high density lidar data was actually developed and published over 20 years ago and about at the same time with ABA method. Check for example Hyyppä, J and Inkinen I. 1999. Detecting and estimating attributes for single trees using laser scanner. Photogramm J Finland 16, 27-42 in comparison with Naesset’s paper published in Remote Sensing of Environment in 1997 and studies published by Nelson et all before that.

Thank you for suggesting the relevant references. The paragraph about ALS was reformulated and references were added. (Lines 58-69)

ALS usually covers areas on regional scale. High density ALS (10 pulses/m2 and more) provides sufficient detail to detect individual trees either from ALS-derived canopy height models [9] or the whole depth of ALS data [10], but typically, parameters of forest stands are derived from ALS data using area-based approach, which considers area as the unit of interest and generally includes estimates of forest variables on stand-level, per hectare values or mean tree attributes [11]. Beyond estimating basic forest parameters such as diameters, height, volume [12–14] biomass or biomass change [15,16], ALS data can be related to leaf area index [17], gap fraction, defoliation [18] or site index [19]. Nowadays, approaches for predicting tree diameter distributions from ALS data emerged [8,20], helping to bridge the gap between area-based and tree-based inventories. On the contrary, TLS provides ultra-high density scans for detailed and accurate reconstruction of a forest stand allowing for deriving virtually any dimension of the forest trees, but produces plot-wise data of low spatial extent; it is time and labor demanding. As shown by numbers of studies, TLS point clouds allow for automatic detection and mapping of forest trees [21], estimating diameters [22] or the complete stem curve and tree architecture [23,24]. Allows to assess accurate non-destructive estimates of aboveground biomass [25] and even temporal development of aboveground biomass [26].

Line 77: “…with survey grade accuracy 10 mm.” Please give reference to this information.

We provide the Riegl VUX data sheet as the reference. Besides, we modified the formulation so that it is in accordance with the specifications: with the presented distance measurement accuracy/precision 10 mm/5 mm (Line 80)

Lines 83-107: Change the order of paragraphs so that at first tell what has been done earlier and after that what is the new thing in this paper and what are the specific aims.

The order of paragraphs was changed according to the suggestion. (Lines 99-113)

Line 152: SI unit symbol is m s-1

The unit symbol was changed from ms-1. to m s-1. (Line 163)

Line 165: ULS data processing is described partly in previous chapter. This chapter is more about point cloud processing.

Yes, point cloud processing is a more precise formulation. The title of the subsection was modified. (Line 176)

2.6. Diameter estimation

Please refine the presentation of the three different circle fitting methods. Describe which tools, libraries, software were used and give brief overall description of every method.

An information about utilized software was added at the end of the subsection: (Lines 299-301)

Functions for all three circle fitting methods were written in the MATLAB environment as the general algorithms briefly described in this section and the mentioned adoptions of the algorithms were applied.

We have also added a brief description of each method: (Lines 224-273)

Hough transform is a method originally developed to detect geometry objects in images, however the pixel-based algorithm can be adopted to apply on point-based data. The Hough transform circle fitting is based on geometrical definition of circle: circle is a set of points with equal distance (i.e. radius) from the center. Therefore, if circles of given radius are drawn around each point belonging to the circle perimeter, all the new circles intersect in the center of the original searched circle. In practical application, a raster is established, where each cell value represents the count of intersects of newly drawn circles with the given raster cell. A peak in the raster represents the center or the original circle. The weak point of the Hough transform is the need for a priori knowledge about circle radius. If radius is unknown, the algorithm must be performed repeatedly with a set of different radii. The Hough transform is reported to be insensitive to noise or incompleteness of data [30]. Utilization of Hough transform for stem diameter estimation in TLS data was indicated by [31], together with detailed explanation of the method.

Random Sample Consensus (RANSAC) is an iterative stochastic method developed to fit a mathematical model in noisy data based on repeated model fitting to random subsamples. For circle fitting, a minimum subsample that defines a circle, i.e. 3 points, is repeatedly randomly selected and the circle is fitted. The quality of each fit is evaluated by the ratio of points within a defined close distance from the fitted circle, so called inliers. Finally, inliers of the best fit are used to fit the final circle. The number of iterations was set to 1.000. Serviceability of RANSAC in forest mensuration was shown by [32] who utilized RANSAC for delineation of tree crowns in ALS data or by [27] for diameter fitting in TLS clouds.

Robust Least Trimmed Squares (RLTS) algorithm is an stochastic iterative method based on least squares criterion applied on a defined portion of smallest residues, which makes it more reliable in case of presence of outliers [33]. Analogous to RANSAC, RTLS iteratively fits circles to random subsamples of 3 points. In each iteration, squared residuals – squared distances of all points to the fitted circle – are calculated and sorted. A defined trim portion h (h > 1/2) of points with the smallest residuals are selected and least square circle fitting is applied on the selected points. The criterion for evaluating the quality of fit is the sum of squared residuals of the least square circle fit. For detailed description of the method, find [33].

Line 218: “Trees presenting a tree” There is probably a mistake her?

Yes, thank you, the mistake was corrected: Points representing a tree (Line 234)

Line 217- (chapter 217): It is not clear how DBH estimate is defined. Is it from observations between 1 and 2 meters above ground? If yes, how you justify that DBH is from this section? Why not use some other section or a curve fitted in sections’ heights and then exact height of 1.3 meters? Or is there something I did not understand?

Yes, we used all points from the section between 1 and 2 meters above ground to fit a circle, and diameter of this circle served as the estimation of the DBH. We are aware that it is quite a large layer and the DBH is not even in the middle of the section.  The explanation is following:

The point density, which was around 40 points per one meter of stem length in average, but sometimes as low as 5 points for the section, did not allow to fit circles in thinner layers. Thinner layers in many cases fail from circle fitting due to insufficient point representation of the circle.

The taper of this 1 m high section usually corresponds to approx. 1 – 1.5 cm, therefore the points should form point belt of approx. thickness of 1.5 cm due to the stem taper along the 1 m long section. The belt thickness is comparable with other sources of point dispersion along the circle perimeter: the LiDAR point accuracy according to RIEGL specification (1.5 cm), the bark structure (1-2 cm differences). Probably the biggest source of error is the co-registration of individual scans form individual flight lines: accurate co-registration usually is problematic in environments with complex structure, as forests. For co-registration we utilize the standard procedure of RIEGL, which usually works well on objects with flat and clearly defined surfaces, such as buildings, roads, etc. The co-registration error may reach several centimeters in forest environment, resulting in inconsistent circle representation.

The error resulting from estimating the DBH from 1-2 m sections (the center of the section is 0.2 m above the breast height) should be systematic (underestimation) and in the level of several millimeters, which is negligible in comparison with other factors.

Lines 266-270: Please provide formulas or references. It is not clear what authors mean by mean error.

Formulas were added to the text, together with symbol explanation. (Lines 306-312)

Line 315: Is it really an exceptionally dense forest stand? I do not consider 450 stems/hectare as very dense.

I agree, the number of stems itself does not imply very dense forest. However, considering the tree dimensions (mean DBH 44.5 cm and mean height 40 m), the stem density exceeds the typical values listed in growth tables for spruce (for given dimension, the density should be 370-380 stems/hectare according to the growth tables). The canopy closure in this stand obstructs a large portion of laser beams to penetrate to the stems. 

Lines 318-321: If a detected tree without a successful diameter estimation are eliminated then the final tree detection rate is the outcome of estimated diameters. Not outcome of detected trees. Right?

Right. Table 2 illustrates counts of trees that were detected, without a knowledge about the subsequent success rate of diameter estimation. Table 3 shows the success rate of estimated diameters (column Measured). The final numbers (Abstract, Conclusions) show how many trees were correctly detected AND allowed estimating the diameter.

Line 353-354: Of all correctly detected spruce trees. Not all spruce trees.

In this context, it means all spruce trees present in the research plots. Specified in the text for better understandability.(Line 402)       

Figure 7: Image quality is very bad. Needs to be improved.

The images were enlarged for better readability, the images were exported with 400 dpi resolution.

Lines 387-409: You provide here results for several statistical tests. Please mention these tests in methods section, too. Also, you could divide the chapter 3.3 Diameter estimation more clearly in presenting the estimation error and then the test results. Presenting test results (and doing the tests) is a good thing. The presentation of the test results could be improved.

Following text was inserted in the method section: (Lines 315-321)

Statistical testing was involved in order to reveal factors influencing the accuracy of diameter estimation. Two-way ANOVA was utilized to investigate the influence of circle fitting method and tree species on absolute diameter errors. We also investigated the influence of point counts available for circle fitting on diameter estimation accuracy; to eliminate the effect of the previously mentioned factors, generalized linear model (GLM) was fitted with two categorical (circle fitting method and tree species) and one quantitative (point counts) predictors. Linear regression models were built to quantify the effect of point counts for each fitting method separately.

The subsection 3.3 was divided into two parts; a subsection 3.4 Factors affecting the accuracy of diameter estimation was separated from the subsection 3.3. (Line 425)

To improve the presentation of the test results, p-values of the results were presented. For linear regression models, the 95% confidence bounds for the slope parameter were presented. The 95% confidence bounds allow not only testing the statistical significance of the trend, but also quantify the influence of the predictor.  (Lines 437-457)

Lines 415-416: The last sentence belongs to discussion part.

The sentence was moved to Discussion and joined with the discussion about limitation and applicability of the method. (Lines 540-553)

Discussion: Add findings of several previous studies about tree detection success rate and diameter estimation (TLS, ALS). While you present very good accuracy for tree detection rate the diameter estimation accuracy is not very convincing. Also, you could discuss is the diameter estimation accuracy good enough for practical use. Furthermore, how do you see your findings of your test case can be generalized? It seems that the forests present quite easy case for the presented method. Is the method applicable in multi species stands, stands with uneven age structure, young stands with much higher stem density…

A paragraph containing findings of several previous studies regarding diameter estimation from point clouds of different origin (ULS, TLS, photogrammetry) with the emphasis on ULS was added to the discussion, comparing our diameter estimation error with their results (Lines 511-526):

The earlier published study of Wieser et al. [27], focusing on diameter estimation from manually delineated tree stems in ULS point cloud, was performed in a deciduous forest during leaf-off season. Thus, their point representation of stem surface was significantly denser. They did not provide RMSE of DBH estimation; however, their errors varied between -18 and 18 cm, with median absolute error, which is always smaller than RMSE, of approx. 5 cm. Here we show that comparable accuracy of DBH estimation can be reached even in coniferous forest with dense canopies and using automated segmentation. Brede et al. [26] report RMSE of diameter estimation 4.24 cm, however, the result is based on fitting 39 trees with the best point representation from the total 58 research trees. Moreover, the suitability for circle fitting was manually inspected for each tree, and outlier points, such as returns from branches, were manually removed. Comparably high RMSE of diameter estimation are reported from RANSAC diameter estimation from TLS data that provides significantly higher point densities and level of detail: Olofsson et al. [28] report RMSE 33 to 59 mm for data without the removal of outliers. RMSE 3 to 6 cm of diameter estimation is reported also from detailed photogrammetric reconstruction of forest stands, eg. [30,36]. The results indicate that for the present we have to accept centimeter-level error for diameter estimation from all kinds of remotely sensed structural data. 

We also discuss the accuracy of diameter estimation and sources of error (Lines 527-539):

The diameter estimation suffers from a relatively large dispersion of points around cross-sections’ perimeters. The point dispersion is partially caused by the diameter change along the 1 m sections of tapered trees. However, the point density, which was around 44 points per one meter of stem length in average, but sometimes as low as 5 points for the section, did not allow to fit circles in thinner layers. Thinner layers in many cases fail from circle fitting due to insufficient point representation of the circle. Due to the taper, the vertical point projection of 1 m long section forms a point belt of the thickness of around 1-2 cm instead of a circle. Another component of the point dispersion is caused by LiDAR accuracy (1-2 cm) and bark structure (1-2 cm). These components represent an unavoidable error of the data acquisition. The most important source of point dispersion is the co-registration of scans from individual flight lines. Due to the limited accuracy of the utilized IMU, software-based co-registration of scans must be applied. Accurate co-registration is problematic in environments with complex structure, as forests. The co-registration error may reach several centimeters in forest environments, resulting in inconsistent and dispersed circle representation.

Finally, a paragraph discussing limitations and applicability of the presented method was added (Lines 540-553):

In a single UAV flight, 10 to 20 hectares of forest area can be covered with LiDAR data. Considering time consumption for mission planning, flight preparation and transports between flight locations, the area that can be mapped in a day reaches 100 hectares. As a result, ULS is effective for detailed mapping at the level of forest stands or forest areas up to several hundreds of hectares. We showed, that for mature even-aged production forests, virtually every tree in the mapped area can be localized with an error that usually did not exceed the tree radius. The accuracy of diameter estimates may be sufficient for common stand inventories as volume assessment before standing timber sale or forest management plan development. An important benefit of USL data is the possibility to measure upper stem diameters to reconstruct the taper curve for wood quality and assortment prediction. The biggest limitation of accurate diameter estimation – centimeter level noise caused by inaccurate co-registration of scan lines – may limit circle detection in point clouds representing thinner trees. Revision of the methods or modified setting may be needed in younger forest stands. The applicability of this method for tree detection and measurement in more complex stands, such as two-storey stands is a subject of further research.

Lines 490-491: “Considering that in a single UAV flight lasting 20 minutes, 10 to 20 hectares of forest area can be covered with high quality point cloud…” Maybe the flight time for data acquisition is about this in optimal conditions, but it requires that everything is ready for the data collection. It is more informative if you give your estimation on how many hectares can be acquired in a single day and how much time is required for planning before the data acquisition and analyzing the data after the data is acquired. Or you just can leave out this kind of marketing text.

Well, this was not meant as a marketing text, but as information about the forest area extent that can be efficiently covered with ULS. The intended meaning was: if you have tens of hectares that you want to map, ULS is the right solution for you; if you have thousands of hectares, consider ALS as better solution, because ULS is not efficient anymore.

We removed this paragraph from conclusions. We reformulated the sentence so that it does not look like marketing text, but discussion about the efficient extent of this method. The modified text was added to discussion (see the comment above) (Lines 540-553).

Line 493: Should it be comparable?

This expression was left out due to the reformulation.

Reviewer 2 Report

Dear authors,

thank you for your very pleasant to read an straight forward article ‘Very high density point clouds from UAV laser scanning for automatic tree stem detection and direct diameter measurement’. To my knowledge this is the first paper using ULS for DBH estimations. The topic is novel and of interest. The methodology seems to be sound and adequate. I have some recommendations to improve the manuscript, which should not avoid publication in any moment.

Best regards,

General remarks

In the introduction you deliver a very valuable overview of SfM and ALS techniques for such forest inventories, therefore I think this paper, estimating DBH values from UAV-SfM could be of interest for your introduction and discussion:

Fritz, Andreas, Teja Kattenborn, and B. Koch. "UAV-based photogrammetric point clouds—Tree stem mapping in open stands in comparison to terrestrial laser scanner point clouds." Int. Arch. Photogramm. Remote Sens. Spat. Inf. Sci 40 (2013): 141-146.

Do you have an precision estimate for the GNSS used for the ‘ground truth’, and have you corrected for the fact that you could not use the GNSS within the tree?

Please add the software or programming environments you used to the manuscript.

I’m quite surprised that your RMSEs are that high, since you work with a very advanced ULS system and your approaches seem to be very sophisticated. Figure 6 is an example were your methods work well. For me as a reader it would be additionally valuable to see an example of a stem were the fitting did nor work that well. Maybe even including circles from all three methods. Could you name some more errors disturbing the fitting in the discussion?

Did I got this correct, that you fitted a circle every full meter along the stem? This would mean there was no circle at breast height. This could explain your small biases.

Minor comments:

I think scots pine is a little more common than scotch pine but I don’t have a strong preference.

218: ‘Trees representing a tree’, Points representing a tree?

411: Shouldn’t the transformation bias from WGS84 to S-JTSK be identical for the UAV-GNSS and the total station?

Author Response

Dear reviewer,

Thank you for your valuable comments. We really appreciate your constructive effort that helped us to eliminate some mistakes in the manuscript, to improve the presentation of the message of our work, and to increase the understandability for readers.

Here your comments are listed together with our response to each comment.

Authors

General remarks

In the introduction you deliver a very valuable overview of SfM and ALS techniques for such forest inventories, therefore I think this paper, estimating DBH values from UAV-SfM could be of interest for your introduction and discussion:

Fritz, Andreas, Teja Kattenborn, and B. Koch. "UAV-based photogrammetric point clouds—Tree stem mapping in open stands in comparison to terrestrial laser scanner point clouds." Int. Arch. Photogramm. Remote Sens. Spat. Inf. Sci 40 (2013): 141-146.

Thank you for the suggestion of the relevant research that we missed. We have added the paper in the introduction (… Similarly, cylinder fitting based on RANSAC was applied on 0.5 thick layers in detailed UAV-acquired photogrammetric point clouds of tree stems…) . (Lines 194-195)

Do you have an precision estimate for the GNSS used for the ‘ground truth’, and have you corrected for the fact that you could not use the GNSS within the tree?

The tree positions were measured using a total station. Neither the total station allows to measure within the tree, therefore the measurement details were added to the manuscript (Lines 143-147):

To record the position of the true center of trees cross-sections the angle offset measuring method was used: the cube corner prism was placed on the side of the tree preserving the distance to the center of the cross-section, and subsequently, the angle to the tree center was recorded separately. The total station data was referenced using two RTK GNSS points. The final error of ground truth tree positions should not exceed 5 cm.

Please add the software or programming environments you used to the manuscript.

All the analyses were carried out in MATLAB 2017R. Information added to the text (Lines 177-179)             :

Once the point clouds were produced, the environment of MATLAB R2017b (The MathWorks Inc., Natick, Massachusetts, USA) was utilized to carry out analyses, point cloud processing, and generate graphic outputs.     

I’m quite surprised that your RMSEs are that high, since you work with a very advanced ULS system and your approaches seem to be very sophisticated. Figure 6 is an example were your methods work well. For me as a reader it would be additionally valuable to see an example of a stem were the fitting did nor work that well. Maybe even including circles from all three methods. Could you name some more errors disturbing the fitting in the discussion?

A paragraph discussing this problem was included to Discussion (Lines 527-539):

The diameter estimation suffers from a relatively large dispersion of points around cross-sections’ perimeters. The point dispersion is partially caused by the diameter change along the 1 m sections of tapered trees. However, the point density, which was around 44 points per one meter of stem length in average, but sometimes as low as 5 points for the section, did not allow to fit circles in thinner layers. Thinner layers in many cases fail from circle fitting due to insufficient point representation of the circle. Due to the taper, the vertical point projection of 1 m long section forms a point belt of the thickness of around 1-2 cm instead of a circle. Another component of the point dispersion is caused by LiDAR accuracy (1-2 cm) and bark structure (1-2 cm). These components represent an unavoidable error of the data acquisition. The most important source of point dispersion is the co-registration of scans from individual flight lines. Due to the limited accuracy of the utilized IMU, software-based co-registration of scans must be applied. Accurate co-registration is problematic in environments with complex structure, as forests. The co-registration error may reach several centimeters in forest environments, resulting in inconsistent and dispersed circle representation.

We added a figure to show an example of point structures representing a good situation with nicely visible stem cross-section and  a more complex situations with incomplete circle and outliers (Figure 8).

Did I got this correct, that you fitted a circle every full meter along the stem? This would mean there was no circle at breast height. This could explain your small biases.

Yes, we fitted a circle every full meter along the stem. The point density, which was around 40 points per one meter of stem length in average, but sometimes as few as 5 points, did not allow to fit circles in thinner layers. As the DBH estimate, we used the section between 1 and 2 meters above ground. The taper of this section usually corresponds to approx. 1 – 1.5 cm, therefore we should have a point belt of approx. thickness of 1.5 cm due to the stem taper along the 1 m long section. This is comparable with other sources of point dispersion along the circle perimeter: the LiDAR point accuracy (1 cm), the bark structure (1 cm differences). Probably the highest source of error is the co-registration of individual scans form individual flight lines: accurate co-registration usually is problematic in environments with complex structure, as forests. For co-registration we utilize the standard procedure of RIEGL, which usually works well on objects with flat and clearly defined surfaces, such as buildings, roads, etc. The co-registration error may reach several centimeters in forest environment, resulting in inconsistent circle representation.

Minor comments:

I think scots pine is a little more common than scotch pine but I don’t have a strong preference.

All occurrences in the text were corrected.

218: ‘Trees representing a tree’, Points representing a tree?

Yes, the mistake was corrected: Points representing a tree (Line 234)

411: Shouldn’t the transformation bias from WGS84 to S-JTSK be identical for the UAV-GNSS and the total station?

Unfortunately, it is not. Due to historical development of the S-JTSK system and its local distortion, there is no universal transformation equation; instead the transformation is carried out using local transformations. The geodetic GNSS receiver provides coordinates in S-JTSK after transformation that utilizes official Czech transformation key provided by state cartographic and surveying institute. This official transformation key is not available for LAS datasets.

Round 2

Reviewer 1 Report

Thank you for improving the manuscript significantly. I have only few minor comments.

Remove “UAV” in keywords. It is mentioned in title.

Line 171: Riegel -> Riegl

Lines 514-515: “…with median absolute error, which is always smaller than RMSE, …” Median absolute error can be the same as RMSE. It is not always smaller.

Lines 538-539: “The co-registration error may reach several centimeters in forest environments, resulting in inconsistent and dispersed circle representation.” Reference missing.

Check references for possible duplicates and or missing refs. “Ayrey, E.; Fraver, S.; Kershaw, J.A.; Kenefic, L.S.; Hayes, D.; Weiskittel, A.R.; Roth, B.E. Layer Stacking: A Novel Algorithm for Individual Forest Tree Segmentation from LiDAR Point Clouds. Can. J. Remote Sens. 2017, 43, 16–27.” is listed twice.
